# FFT-DM: A Decomposable Forward Process in Diffusion Models for Time-Series Forecasting

## Abstract

We introduce FFT-DM (Fast Fourier Transform–Diffusion Model), a model-agnostic forward diffusion process for time-series forecasting that decomposes signals into spectral components, preserving structured temporal patterns such as seasonality more effectively than standard diffusion. Unlike prior work that modifies the network architecture or diffuses directly in the frequency domain, FFT-DM alters only the diffusion process itself, making it compatible with existing diffusion backbones (e.g., DiffWave, TimeGrad, CSDI). By staging noise injection according to component energy, FFT-DM maintains high signal-to-noise ratios for dominant frequencies throughout the diffusion trajectory, thereby improving the recoverability of long-term patterns. This strategy enables the model to maintain the signal structure for a longer period in the forward process, leading to improved forecast quality. Across standard forecasting benchmarks—including Electricity, ETTm1, Temperature, PTB-XL, and MuJoCo, FFT-DM consistently outperforms strong diffusion baselines and rivals state-of-the-art models such as Sashimi, with negligible computational overhead ( 7%). [1]

## 1 Introduction

Time-series forecasting plays a critical role in a wide range of real-world applications (Lim & Zohren, 2021), from weather prediction (Choi et al., 2023) to financial market analysis (Makridakis et al., 2024). Classical statistical methods (Box et al., 2016) and autoregressive linear models (Hyndman & Athanasopoulos, 2021) remain widely used due to their interpretability and competitive predictive performance. However, these approaches often struggle with capturing long-range dependencies (Zhou et al., 2021), hierarchical seasonal structures (Hewamalage et al., 2021), and non-linear dynamics (Rangapuram et al., 2018). Diffusion models (Sohl-Dickstein et al., 2015), which learn a Markov chain of progressively denoised latent representations, have demonstrated state-of-the-art performance in image generation (Ho et al., 2020; Dhariwal & Nichol, 2021) and inpainting (Lugmayr et al., 2022). More recently, they have been applied to time-series forecasting (Kong et al., 2021; Tashiro et al., 2021), showing competitive results against other generative models. While some works have incorporated multi-resolution processing (Shen et al., 2024; Gu et al., 2023a; Fan et al., 2024), yet these models do not fully exploit the inherent structure of time-series data in their diffusion processes.

A well-established technique in time-series modeling is decomposition into three main components: trend, seasonal effects, and residual noise (Hyndman & Athanasopoulos, 2021; Cleveland et al., 1990; Yi et al., 2023). However, existing diffusion models lack mechanisms to preserve these structured components, often leading to inaccurate long-term predictions. For instance, in electricity demand forecasting, standard diffusion-based models tend to smooth out periodic consumption cycles, failing to accurately capture sub-daily patterns. This loss of structured information diminishes the model's forecasting ability, especially in applications that require precise seasonality preservation.

To address this limitation, we introduce a novel approach that integrates structured time-series decomposition into the forward diffusion process. Specifically, we employ Fourier decomposition to isolate and process seasonal components and remainders separately. After identifying the different components, our proposed diffusion process performs diffusion sequentially and in ascending order of amplitude, guaranteeing that the more relevant components are kept longer in the forward diffusion

---

[1]The code for the paper is available at `https://anonymous.4open.science/r/FFT-DM-8E36`.

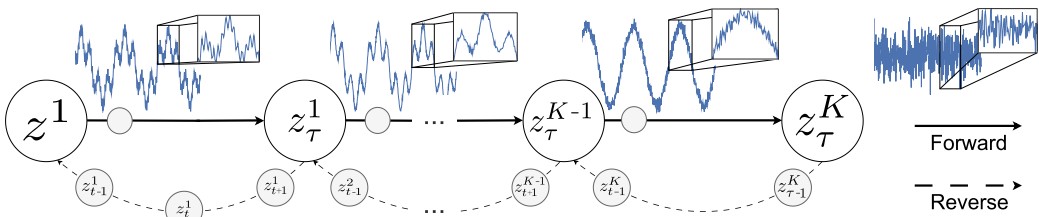

Figure 1: Diffusion process with decomposition. During the forward process, noise is added to each component in ascending order of amplitude. This process slows down the destruction of the more relevant frequencies, preserving key structures longer.

process and reconstructed earlier. Our method modifies the sample generation of the training process, and equivalently the sample inference process, which means that it can be used with any known deep learning architecture for time-series model. This approach is particularly suited for long time-series, where stronger seasonal patterns are common and can be more effectively captured through seasonal decomposition during both training and inference.

Furthermore, we generalize this framework to accommodate any additive decomposition method and demonstrate its efficacy compared to diffusion-based forecasting models that use the conventional forward process.

**Contributions.** Existing diffusion-based time series models (e.g., DiffWave (Kong et al., 2021), TimeGrad (Rasul et al., 2021), MG-TSD (Fan et al., 2024), mrDiff (Shen et al., 2024)) focus primarily on adapting architectural components, such as attention or S4 layers, or developing hierarchical models, but retain the standard forward process that injects noise indiscriminately, erasing structured temporal features such as seasonality and trend. In contrast, we redesign the forward process itself to explicitly preserve these features, yielding more faithful reconstructions and adapted to long-horizon forecasting when structure-aware extrapolation is required. Unlike previous works, our approach is model agnostic and can be applied to any diffusion model architecture.

Our contributions are threefold: (1) we propose a novel structured forward diffusion process leveraging Fourier decomposition for time-series forecasting; (2) we theoretically generalize this framework to accommodate various additive decomposition methods; and (3) we empirically validate our Fourier decomposition approach on multiple benchmark datasets, demonstrating its effectiveness in capturing long-range dependencies and seasonal patterns.

## 2 RELATED WORK

**a) i. Diffusion Models** Diffusion probabilistic models (Sohl-Dickstein et al., 2015; Song et al., 2021) have gained significant popularity since their introduction, demonstrating remarkable success in image generation (Ho et al., 2020; 2022a) and text-to-speech synthesis (Chen et al., 2021). They have also been extended to video (Ho et al., 2022b), audio (Chen et al., 2020), and other applications (Luo & Hu, 2021). Song et al. (2021) unified two perspectives on these models—discrete and continuous formulations—under the umbrella of score-based generative models. In this work, we adopt the discrete formulation, equivalent to the variance-preserving stochastic differential equation (VP-SDE) Ho et al. (2020), in a non-autoregressive setting.

**a) ii. Latent and Multi-scale Diffusion** Recent advancements have improved diffusion models in terms of scalability and efficiency. Techniques such as inference cost reduction (Song et al., 2021), and others (Song & Ermon, 2020; Bortoli et al., 2021; Watson et al., 2021) have enhanced their applicability. Additionally, Nichol & Dhariwal (2021) introduced variance prediction in the reverse diffusion process, enabling competitive performance with fewer sampling steps. For high-resolution image generation, latent diffusion models (Ramesh et al., 2022) compress data into a lower-dimensional space before applying the diffusion process. A different approach applies diffusion in the spectral domain (Phillips et al., 2022; Crabbé et al., 2024). Other hierarchical models refine outputs by generating a low-resolution image first, followed by super-resolution techniques (Gu et al., 2023a; Ho et al., 2022a). However, these improvements remain largely focused on spatial data, and

their direct applicability to time-series forecasting remains limited due to differences in structural dependencies.

**b) i. Temporal Representation** Recent research has focused on improving temporal representations in Transformer-based forecasting models. Fedformer (Zhou et al., 2022) maps time-series data into the frequency domain, selecting a random subspace for modeling, while Pyraformer (Liu et al., 2022) introduces pyramidal attention to enable multi-resolution analysis. Basis expansion techniques, such as N-Beats (Oreshkin et al., 2020), decompose time-series data into trend and seasonal components, with subsequent refinements like N-Hits (Challu et al., 2023), Zeng et al. (2023) and Fons et al. (2024) improving multi-scale modeling and simplifying decomposable time-series structures into linear models. In addition, frequency-aware models like Wu et al. (2023); Lin et al. (2024), and Wang et al. (2024b;a) have presented new layers that are specifically tailored to enhance the ability to extract temporal and multi-seasonal information using Fourier Decomposition to extract top-$k$ frequencies. The first TimeMixer paper (Wang et al., 2024b) hinted during ablation studies that FFT based decomposition yielded better results than moving-average decomposition, while the latter expanded on the use of Fourier decomposition alongside MixerBlocks and dual-axis attention.

**b) ii. Diffusion Time-series** The application of diffusion models to time-series forecasting began with WaveNet (van den Oord et al., 2016) and latter DiffWave (Kong et al., 2021), originally designed for audio and speech generation. Building on DiffWave, TimeGrad (Rasul et al., 2021) applied a similar architecture to diffusion-based forecasting using an auto-regressive setting and Langevin dynamics. Around the same time, ScoreGrad (Yan et al., 2021) introduced a stochastic differential equation (SDE) formulation for time-series diffusion. Subsequent works have proposed incremental improvements, including Transformer-based architectures (Tashiro et al., 2021), structured state-space models (Alcaraz et al., 2022), and multi-resolution forecasting via future mixup (Shen & Kwok, 2023; Shen et al., 2024). The latter employs a multi-stage approach to inference, where different models process varying levels of data granularity. However, this methods rely on fixed-stage decomposition, which can lead to information loss in highly dynamic time-series data. Additionally, none of these approaches incorporate structured time-series decomposition directly into the diffusion process, limiting their ability to preserve long-range dependencies and periodic patterns.

**This paper.** While prior work has made progress in diffusion-based time-series forecasting, it has not addressed the fundamental issue of structured information loss in the forward diffusion process. In contrast, our approach explicitly integrates structured time-series decomposition within the diffusion framework. By leveraging Fourier decomposition, we ensure that periodic components are preserved throughout the forward process, improving forecasting accuracy for seasonal and long-horizon time-series. Unlike prior multi-resolution methods, our approach is fully generalizable to different additive decomposition techniques and does not impose fixed granularity constraints.

chaUnlike prior approaches that operate solely through model architecture or conditioning (e.g., Transformer-based or Cascaded models), our contribution introduces a structured forward diffusion process that incorporates signal decomposition, enabling frequency-aware denoising and better temporal structure retention (Table 1).

Table 1: Feature comparison of FFT-DM with prior diffusion-based time-series models

| Feature | DiffWave Kong et al. (2021) | TS-Diffusion Yuan & Qiao (2024) | TimeGrad Rasul et al. (2021) | CSDI Tashiro et al. (2021) | MG-TSD Fan et al. (2024) | mrDiff Shen et al. (2024) | **FFT-DM (Ours)** |
|---|---|---|---|---|---|---|---|
| Decomposition modules | | | | | ✓ | ✓ | ✓ |
| Frequency awareness | | ✓ | | | ✓ | ✓ | ✓ |
| Architecture agnostic | | | | | | | ✓ |
| Non-autoregressive | ✓ | ✓ | | ✓ | | ✓ | ✓ |
| Seasonal/trend structure | | ✓ | | | ✓ | ✓ | ✓ |
| Forecasting | | | ✓ | ✓ | ✓ | ✓ | ✓ |

## 3 DIFFUSION MODELS

**Diffusion Forward Process**   A diffusion process (Sohl-Dickstein et al., 2015) consists of progressively adding Gaussian noise to a sample $x_0$ over $T$ time-steps, such that the data distribution transitions to an isotropic Gaussian distribution. The process state transition is defined as

$$q(x_t|x_{t-1}) = \mathcal{N}(x_t|\sqrt{1-\beta_t}x_{t-1}, \beta_t \mathbf{I}), \tag{1}$$

or, equivalently,

$$x_t = \sqrt{1-\beta_t}x_{t-1} + \sqrt{\beta_t}\epsilon, \tag{2}$$

where $\epsilon \sim \mathcal{N}(0, \mathbf{I})$, and $\beta_t$ is a noise scheduling parameter controlling the rate of information loss. The values of $\beta_t \in ]0, 1[$ are defined according to a scheduler, such as a linear or cosine schedule (Ho et al., 2020), and follow the constraint $\beta_1 < \beta_2 < \cdots < \beta_T$.

Given independent noise at each time-step $t$, the closed-form expression for $x_t$ at any arbitrary $t$ is

$$x_t = \sqrt{\bar{\alpha}_t}x_0 + \sqrt{1-\bar{\alpha}_t}\epsilon, \tag{3}$$

where $\alpha_t = (1 - \beta_t)$ and $\bar{\alpha}_t = \prod_{i=1}^t \alpha_i$ and $q(x_t|x_0) = \prod_{i=1}^t q(x_i|x_{i-1})$. As $T \to \infty$, the data distribution converges to a Gaussian prior

$$q(x_T|x_0) = q(x_T) = \mathcal{N}(0, \mathbf{I}), \tag{4}$$

that is independent of $x_0$.

This formulation defines the forward diffusion process, which is the focus of this work. In the following sections, we adapt these equations to the case where $x$ has a known lossless decomposition.

**Reverse Diffusion Process**   The goal of diffusion models is to learn a denoising function that approximates the reverse process $q(x_{t-1}|x_t)$. Since this transition is intractable (Bishop & Bishop, 2024), an approximation is used:

$$p(x_{t-1}|x_t, \Theta) = \mathcal{N}(x_{t-1}|\mu_\theta(x_t, t), \sigma_t \mathbf{I}), \tag{5}$$

where $\Theta$ represents the parameters of the neural network, which learns to estimate the mean function $\mu_\theta(x_t, t)$. Under the assumption that $\beta_t \ll 1$, the true posterior $q(x_{t-1}|x_t, x_0)$ is well-approximated by a Gaussian distribution (Sohl-Dickstein et al., 2015). The variance term $\sigma_t$ is often assumed to be known and fixed, although some models also learn it as part of the diffusion process (Nichol & Dhariwal, 2021).

**Training Objective**   The model is typically trained by minimizing the Evidence Lower BOund (ELBO) (Kingma & Welling, 2014), which can be formulated in multiple ways: as a data prediction task, a noise estimation problem, or a step-wise reconstruction objective. The simplified objective is derived from the sum of Kullback-Leibler (KL) divergences:

$$L = \sum_{t=2}^T KL(q(x_{t-1}|x_t, x_0)||p(x_{t-1}|x_t, \Theta)) \tag{6}$$

$$L = \sum_{t=2}^T \|m(x_t, x_0) - \mu(x_t, \Theta, t)\|^2 + c. \tag{7}$$

Another approach involves rewriting $\mu(x_t, \Theta, t)$ as a function of a noise model (Luo, 2022):

$$\mu(x_t, \Theta, t) = \frac{1}{\sqrt{\alpha_t}}\left(x_t - \frac{\beta_t}{\sqrt{1-\bar{\alpha}_t}}\right)\epsilon(x_t, \Theta, t). \tag{8}$$

In this case the noise predicting model $\epsilon(x_t, \Theta, t)$ leads to the simplified loss

$$L_{\epsilon t} = \|\epsilon - \epsilon(x_t, \Theta, t)\|^2$$

.

This formulation allows predicting the total noise added to the data using equation 3, a technique successfully applied in time-series diffusion models (Alcaraz et al., 2022; Tashiro et al., 2021).

**Conditional Diffusion for Time-Series Forecasting**  In time-series forecasting, a special case of imputation, the goal is to predict future values $\mathbf{x}_0^{P+1:W} \in \mathbb{R}^{d \times (W-P)}$ given past observations $\mathbf{x}_0^{0:P} \in \mathbb{R}^{d \times P}$. Here, $W - P$ is the length of the forecast window, and $P$ is the length of the lookback window. In the most general case, let $\mathbf{x}_0^{Obs}$ denote the observed time-steps and $\mathbf{x}_0^{Tgt}$ the target time-steps.

In conditional diffusion models, the denoising process at step $t$ is given by (Shen et al., 2024):

$$p_\theta(\mathbf{x}_{t-1}^{Tgt}|\mathbf{x}_t^{Tgt}, \mathbf{c}) = \mathcal{N}(\mathbf{x}_{t-1}^{Tgt}; \mu_\theta(\mathbf{x}_t^{Tgt}, \mathbf{c}, t), \beta_t \mathbf{I}), \tag{9}$$

where $\mathbf{c} = \mathcal{F}(\mathbf{x}_0^{Obs}, t)$ represents the conditioning information extracted from past observations. The transformation $\mathcal{F}$ encodes relevant features such as past time-series values and temporal embeddings. For simplicity, in the remainder of this paper, we omit notation explicitly indicating this conditioning when it is clear that the model is conditioned on observable time-steps.

## 4 FOURIER DECOMPOSITION DIFFUSION MODEL

**Forward Process**  We propose FFT-DM, a model-agnostic modification of the diffusion forward process for time-series forecasting. In standard diffusion models, noise is applied indiscriminately to the entire input (Ho et al., 2022a), so every diffusion step simultaneously corrupts all aspects of the signal. In contrast, many time-series can be decomposed into disjoint spectral components (e.g., seasonalities and residuals) whose sum reconstructs the original signal. FFT-DM leverages this property by staging noise injection across components, so that the reverse diffusion process focuses on reconstructing the dominant structures first before addressing finer residuals. This sequential decomposition does not add learning complexity, preserves periodic patterns longer into the diffusion trajectory, and can be applied to any backbone without architectural changes.

To formalize this, we introduce a **generalized forward diffusion process** that incorporates signal decomposition and provides a closed-form expression for each forward step at time $t$. First consider that a sample $x_0$ is expressed as the sum of its decomposed components:

$$x_0 = z_0 = \sum_{k=1}^{K} f_0^k \tag{10}$$

where $D$ is the operator that linearly decomposes the sample $z_0$ into $K$ components $\{f_i\}_{i=1}^{K}$. These components are, by definition, orthogonal.

Considering that $\{z_t^k\}_{t=1,\dots,\tau}^{k=1,\dots,K}$ is the set of latent variables that will be transformed from the initial data $z_0$ to a gaussian distribution, the initial step $q(z_1^1|z_0)$ is defined as:

$$z_1^1 = \sqrt{1 - \beta_1} f_0^1 + \sqrt{d_1 \beta_1} \epsilon + \sum_{k=2}^{K} f^k \tag{11}$$

where $\{\beta_i\}_{i=1}^{\tau}$, is the noise scheduling parameter of the staged diffusion, from $i$ to $\tau$. Note that, since the staged diffusion runs only for $\tau < T$ steps, the noise schedule parameters $\beta_{t=1}^{\tau}$ and their cumulative products $\bar{\alpha}_t = \prod_{t=1}^{i}(1 - \beta_t)$ are defined with respect to this shortened horizon. The first intermediate step $q(z_\tau^1|z_{\tau-1}^1)$:

$$z_\tau^1 = \sqrt{1 - \beta_\tau} f_{\tau-1}^1 + \sqrt{d_1 \beta_\tau} \epsilon + \sum_{k=2}^{K} f^k \tag{12}$$

$$z_\tau^1 = f_\tau^1 + \sum_{k=2}^{K} f^k = z_0^2 \tag{13}$$

and $q(z_1^2|z_0^2)$:

$$z_1^2 = \sqrt{1 - \beta_\tau} f_0^2 + \sqrt{d_2 \beta_\tau} \epsilon + \sum_{k=3}^{K} f^k + f_\tau^1 \tag{14}$$

Here, $K$ represents the number of significant components, and always contains a component that corresponds to the residual term. This ensures a **lossless decomposition**, and trivially the usual standard diffusion process occurs when $K = 1$. Finally, T is the total number of time steps, and $\tau = \frac{T}{K}$, is the number of time-steps for each component.

To regulate noise diffusion, we introduce a **signal-to-noise ratio (SNR) scaling factor** $d_k$ (Oppenheim et al., 1996). This factor adjusts the noise added to each frequency component, preventing excessive diffusion in the early stages and ensuring that noise does not spill over into subsequent components (Fig. 2). In Annex C we evaluate in detail the SNR of our proposed diffusion process. This idea has been explored in hierarchical and cascading diffusion models (Gu et al., 2023b;a).

Applying equation 2 to $f^k$ with the SNR, meaning that when $t = \tau$, $f_\tau^k \sim N(0, d_k I)$ we derive:

$$f_t^k = \sqrt{1 - \beta_t} f_{t-1}^k + \sqrt{d_k \beta_t} \epsilon, \tag{15}$$

and

$$z_t^k = f_t^k + \sum_{n>k}^{K} f_0^n + \sqrt{\sum_{n=1}^{k-1} d_n} \epsilon. \tag{16}$$

This leads to the closed-form expression for $q(z_t^k | z_0)$:

$$z_t^k = \sqrt{\bar{\alpha}_t} f_0^k + \sum_{n>k}^{K} f_0^n + \sqrt{d_k(1 - \bar{\alpha}_t)} \epsilon + \sqrt{\sum_{n=1}^{k-1} d_n} \epsilon$$

$$z_t^k = \sqrt{\bar{\alpha}_t} f_0^k + \sum_{n>k}^{K} f_0^n + \sqrt{-d_k \bar{\alpha}_t + \sum_{n=1}^{k} d_n} \epsilon, \tag{17}$$

for diffusion step $t$ in stage $k$, with $\epsilon \sim N(0, I)$. The term $\sum_{n>k}^{K} f_0^n$ represents the frequencies/components yet to be diffused, while the frequencies already diffused are defined as $\sum_{n=1}^{k-1} d_k$, in the first line of equation 17, using the property of the variance of the sum of independent variables. note also that, as can be inferred from equation 13, $z_\tau^k = z_0^{k+1}$ for $k < K$ and $z_\tau^K \sim N(0, I)$

The **SNR scaling factor** ($d_k$) for each component is computed as:

$$\text{SNR} = \frac{\mathbb{E}[(f^k)^2]}{\mathbb{E}[\epsilon^2]}, \tag{18}$$

ensuring that noise diffusion is proportional to the amplitude of each frequency component. We highlight that this scaling factor is calculated based on the components, and therefore is not a hyperparameter. In Fig. 2 we empirically show that SNR rescaling is necessary to guarantee that each component is diffused during its respective period. While we validate the decomposition approach using Fourier decomposition, explained in detail in Annex A.1, the diffusion process itself remains **agnostic to the specific decomposition method**, as long as it satisfies the weak assumption of lossless reconstruction.

**Reverse Process**  As we are applying a sample specific decomposition with Signal-to-noise-ratio $d_k$, which might not be constant, during the inference process this value is unknown. One possible solution is to consider $d_k$ as part of the output of the model, modifying the Loss function in equation 8. However, in this case, we estimate these values from the conditioning time-steps during inference. This is equivalent to performing the decomposition on the masked samples, however in this case we are only interested on the amplitude of the component, and not the component itself.

Using equation 17, we use the reparametrization trick and reformulate the problem as

$$f_0^k = \frac{z_t^k - \sqrt{-d_k \bar{\alpha}_t + \sum_{n=0}^{k} d_n} \epsilon - \sum_{n>k}^{K} f_0^n}{\sqrt{\bar{\alpha}_t}}$$

$$f_0^k = \frac{z_t^k - \epsilon' - \sum_{n>k}^{K} f_0^n}{\sqrt{\bar{\alpha}_t}}, \tag{19}$$

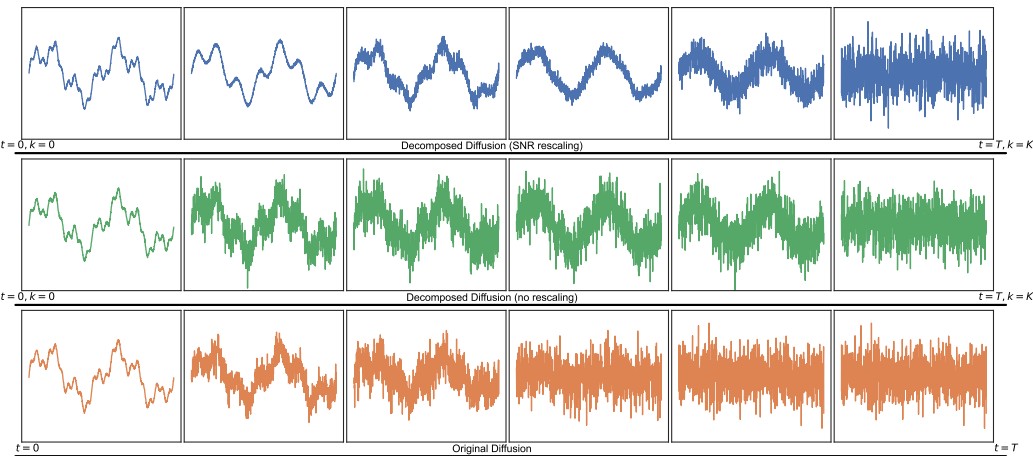

Figure 2: Effect of SNR scaling in noise application. Key frequencies in the signal remain discernible for a longer duration. Without SNR scaling, noise spreads across components, degrading separability. All approaches in this figure use a linear schedule $0.002 < \beta < 0.02$.

where $\epsilon(t, k)' = \sqrt{-d_k \bar{\alpha}_t + \sum_{n=0}^{k} d_n} \epsilon$. This modifies the Simplified Loss (Ho et al., 2020) term to be

$$L_{\epsilon t} = \|\epsilon'(t, k) - \boldsymbol{\epsilon}'(x_t, \Theta, t, k)\|^2. \tag{20}$$

It is straightforward that $(t, k)$ are already inputs to the model, and therefore the only significant change is that the model outputs an already scaled version of $\epsilon$ conditioned on $x_t, t, k$, which means that for lower $k$ and $t$, the variance of the scaled noise is smaller. Using this simplified loss, we incur on an implicit weighted loss that will give more importance to the last steps of the forward diffusion process.

The inference procedure, then becomes $p(z_{t-1}^k | z_t^k, \Theta) \sim N(z_{t-1}^k | \mu(\mathbf{z}_t^k, \Theta, t, k), \sigma(\alpha, \hat{d}_k))$:

$$\mu(\mathbf{z}_t^k, \Theta, t, k) = \frac{1}{\sqrt{\alpha_t}} \left( \mathbf{f}_t^k - \frac{1 - \alpha_t}{1 - \bar{\alpha}_t} \boldsymbol{\epsilon}_\theta'(\mathbf{z}_t^k, t, k) \right) + \sum_{n>k}^{K} \mathbf{f}_0^n \tag{21}$$

where $\mathbf{f}_t^k$ is the estimated component at each time-step, $f_0^n$ for $n > k$ are the already inferred components, and the scale component $\sigma_t^k$ is defined as:

$$\sigma_t^k = \frac{d_k(1 - \alpha_t)(1 - \bar{\alpha}_{t-1})}{1 - \bar{\alpha}_t}$$

which is similar to the original formulation, multiplied by the SNR term.

Since the loss function and estimated noise have changed, compared with the traditional diffusion process, the sampling process needs to be modified accordingly.

**Sampling** The sampling procedure follows the logic of the decomposition, effectively predicting each component in descending order. The main difference in this procedure is the need to calculate the SNR parameter $d_k$. In this work, we present the case where $d_k$ is sample specific, since time-series samples are not homogeneous, and have different frequencies, with different amplitudes and even different number of significant frequencies, but depending on the dataset, it can be defined as a batch average, a total average, or an input hyperparameter. The derivation of the reverse diffusion equations is presented in Annex B.

**Algorithm 1** Training

---

**repeat**

$z_0 \sim q(x_0)$

$t \sim \text{Uniform}(0, T)$

$F = \text{Decomp}(x_0, K)$

$k \sim \text{Uniform}(0, K)$

$z_t^k = \sqrt{\bar{\alpha}_t} f_0^k + \sum_{n>k}^K f_0^n + \sqrt{-d_k \bar{\alpha}_t + \sum_{n=0}^k d_n} \epsilon$

$\epsilon' = \sqrt{-d_k \bar{\alpha}_t + \sum_{n=0}^k d_n} \epsilon$

Take gradient descent step on

$$\nabla_\theta \left\| \epsilon' - \epsilon_\theta \left( z_k^t, t, k \right) \right\|^2$$

**until** convergence

---

**Algorithm 2** Inference

---

**repeat**

$\mathbf{z}_T^K \sim N(0, 1)$

$\hat{F}, \hat{d}_k = D(x_0^{Obs}, K)$

**for** $k = K, .., 1$ **do**

  **for** $t = T, ..., 0$ **do**

    **if** $t > 0$ **then**

      $\mathbf{z}_{t-1}^k = update(\mathbf{z}_t^k, \sum_{n>k}^K \mathbf{f}_0^n)$ equation 21

      $\mathbf{z}_{t-1}^k = \mathbf{z}_{t-1}^k + \sqrt{\hat{d}_k \sigma_t + \sum_{n=0}^{k-1} d_n} \epsilon$

    **else**

      $\mathbf{z} = update(\mathbf{z}_1^k, \sum_{n>k}^K \mathbf{f}_0^n)$ equation 21

    **end if**

  **end for**

**end for**

**until** finished

---

The Algorithm 2 indicates the inference procedure to produce a sample using the trained model.

## 5 MODELS

To evaluate and compare our proposed forward diffusion process, we employed established non-auto-regressive time series conditional diffusion models. All these models are non-auto-regressive to improve computation time, and use masking-based conditioning, which generalizes forecasting as a specific type of imprinting. In this section, we present a quick overview of the models, its overall architecture, and the only modification applied to the models, to accommodate the stage steps.

**DiffWave** (Kong et al., 2021) is a conditional generative model designed primarily for tasks such as speech synthesis, with non-auto-regressive forecasting. Its architecture employs a 1D convolution neural network with residual blocks. This specific architecture has been used in other works that present different models using the same architecture, such as WaveNet (van den Oord et al., 2016), which is a generative network for waveform generation, and ScoreGrad (Yan et al., 2021) which uses the energy score matching formulation for diffusion models. With the exception of the Sashimi model, all other models have similar architectures, and the hyperparameters are present in full in Annex A.5. **CSDI** Tashiro et al. (2021) is a diffusion model for time series imputation and forecasting that changes upon the DiffWave model by using a 2-D attention mechanism in each residual layer instead of a convolution layer. The **SSSD** Alcaraz et al. (2022) model replaces the bidirectional convolutional layers in DiffWave with a structured state space model (S4) (Gu et al., 2022). **Sashimi** is an auto-regressive U-net architecture (Goel et al., 2022) with a starting size of 128 and with a downsampling and feature expansion factor of 2. The residual connections are defined by an S4 layer and linear layers at each pooling level. In here we use the adaptation in (Gu et al., 2022) to conditional generation in a non-autoregressive setting.

## 6 RESULTS

In this section, we present the results for forecasting across multiple datasets and compare each model with and without our forward diffusion process. All experiments use the complete lookback window as part of the conditional features, and the complete hyperparameters are presented in A.2. All models were trained locally, to guarantee consistency of results and methods, in a cluster environment with 4 A100 Nvidia, and to ensure replicability all hyperparameters used are the ones identified by the authors of the original models, when available.

In Table 2 we observe that adding the forward decomposition method enhances the results in most cases. Taking into account that some datasets have more seasonal features than others, we expected to see the biggest effect on data with strong seasonal effects. The Electricity and PTB-XL datasets have clearly defined seasonalities that can be identified in separate components, in particular in the electricity dataset, which has a clearly defined daily fluctuation. For most models, the FFT

Table 2: Performance comparison of forecasting with state-of-the-art models with and without the Fourier Decomposition forward process.

| Model | PTB-XL | | MUJOCO | | Electricity | | ETTm1 | | Temperature | | Solar | |
|---|---|---|---|---|---|---|---|---|---|---|---|---|
| | MSE | MAE | MSE | MAE | MSE | MAE | MSE | MAE | MSE | MAE | MSE | MAE |
| DiffWave | 0.081 | 0.150 | 0.0583 | 0.183 | 2.564 | 0.7132 | **1.0932** | **0.7724** | 0.0035 | 0.029 | 0.444 | 0.32 |
| + FFT-DM (Ours) | **0.066** | **0.139** | **0.0021** | **0.0265** | **1.31** | **0.54** | 1.7743 | 1.0403 | **0.0024** | **0.025** | **0.354** | **0.30** |
| SSSD | 0.064 | 0.111 | 0.0058 | 0.051 | **0.957** | **0.300** | 1.642 | 0.9190 | 0.0025 | 0.023 | 0.568 | 0.37 |
| + FFT-DM (Ours) | **0.0584** | **0.0812** | **0.00062** | **0.0093** | 1.283 | 0.750 | **1.398** | **0.8845** | **0.0024** | 0.023 | **0.505** | **0.36** |
| Sashimi | **0.056** | **0.099** | **0.0007** | **0.0103** | 2.187 | 0.4547 | 1.2070 | 0.788 | 0.0030 | 0.027 | 0.487 | 0.35 |
| + FFT-DM (Ours) | 0.0720 | 0.152 | 0.0008 | 0.0232 | **1.295** | **0.5549** | **0.8705** | **0.678** | **0.0024** | **0.023** | **0.448** | **0.33** |
| CSDI | **0.078** | 0.123 | 0.0025 | 0.032 | 1.314 | **0.532** | 1.123 | 0.790 | 0.0031 | 0.023 | **0.472** | 0.35 |
| + FFT-DM (Ours) | 0.079 | **0.091** | **0.0013** | **0.021** | **1.180** | 0.641 | **0.73244** | **0.667** | **0.0024** | **0.022** | 0.480 | 0.35 |

decomposition provides an improvement upon the mean squared error of the forecast. On the other hand, datasets like MuJoCo have little to no trend or seasonality, and the improvements we identify in Table 2 are mostly related to the slower noising process that is a byproduct of this method.

While the authors understand that forecasting is not a task usually performed for ECG data, we include this dataset for comparability with prior diffusion-based works that used PTB-XL in a forecasting setup (Tashiro et al., 2021; Alcaraz et al., 2022). The PTB-XL dataset has the longest samples, so the estimation of the signal-to-noise ratio for each component during inference is more precise. For this task, the forecast window comprises roughly a complete cardiac cycle (Fig. 7, Annex G). In this example, we observe that the cardiac peaks are well estimated, particularly for the SSSD and Sashimi models with FFT. Also, we note that the Sashimi model, which is a U-net architecture, seems to be the model that benefits the least from this formulation, while CSDI and DiffWave consistently improve when using our decomposition.

**Limitations.** While FFT-DM improves time-series forecasting, it has some limitations. First, it assumes periodic components in the data, making it less effective for highly non-stationary signals. Second, the approach introduces additional one tunable hyperparameter, the number of components, requiring empirical tuning for each use case. Additionally, estimating SNR at inference time assumes past signal structure remains representative, and that there is enough conditional data to correctly estimate, which may not always hold. Finally, performance varies across datasets, with greater benefits observed in data with strong seasonal patterns. Future work should explore adaptive decompositions and automated parameter selection.

## CONCLUSIONS AND FUTURE WORK

In addition to improved forecasting accuracy, FFT-DM provides a modular mechanism that structures the forward diffusion process, helping to preserve salient temporal patterns. This design offers a degree of interpretability, through explicit control of how components degrade, that may be valuable when structure preservation is important. Although our experiments focused on standard forecasting benchmarks, the approach suggests potential in application areas such as energy systems, health monitoring, and finance, where explainability and reliable long-term behavior are desirable.

Our experiments show that the proposed forward process consistently outperforms state-of-the-art diffusion models for time-series forecasting under standard training and inference procedures. An ablation study on the number of components confirms that using an appropriate decomposition improves results and guides the design of the stage-wise noise schedule. Since diffusion for forecasting is still relatively new, future work may explore architectures and hyperparameters more specifically tailored to this task. Because FFT-DM is decomposer-agnostic, different time-series domains could benefit from bespoke decompositions informed by domain expertise. Finally, extending scheduler optimization to jointly tune both the diffusion rate ($\beta$) and stage durations ($\tau$) is a promising direction for improving efficiency and accuracy.

## REPRODUCIBILITY

We have made significant efforts to ensure the reproducibility of our results. All details of the proposed methodology, including model architectures, training procedures, and hyperparameter settings, are described in Section 5 of the main paper and further expanded in Appendix A.2. The datasets used in our experiments are publicly available. For theoretical results, all assumptions and proofs are included in Appendix B. To facilitate replication of our experiments, we provide anonymized source code `https://anonymous.4open.science/r/FFT-DM-19E8` and scripts, which reproduce the training and evaluation pipelines end-to-end.

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

## A  ARCHITECTURE AND DATA DETAILS.

### A.1  FOURIER DECOMPOSITION

**Fourier Decomposition**  Fourier decomposition is a fundamental signal processing technique that breaks down a time series into a sum of sinusoidal components, each linked to a specific frequency. Formally, a time series $\{x^0, x^1, x^2, ...\}$ can be expressed as:

$$x^j = \frac{1}{P} \sum_{w=0}^{P-1} X_w e^{i2\pi \frac{w}{P} j}, \tag{22}$$

where $X_w$ are the Fourier coefficients, $P$ is the period of the time series, and $w$ is the frequency index. These coefficients are computed as:

$$X_w = \sum_{j=0}^{P-1} x^j e^{-i2\pi \frac{w}{P} j}. \tag{23}$$

According to the **Fourier Inversion Theorem** Oppenheim et al. (1996), any sufficiently smooth signal can be uniquely reconstructed using the inverse Fourier transform as $T \to \infty$, with the Fourier coefficients diminishing in magnitude for higher frequencies Fourier (1822). In our forward diffusion process, we use Fourier decomposition to isolate **distinct frequency components**, leveraging the **linear separability of the Fourier Transform** Hyndman & Athanasopoulos (2021). This allows the model to learn the reverse diffusion process separately for each dominant frequency, with a final stage dedicated to residuals, ensuring **lossless decomposition**.

**Frequency Identification**    To identify the most important frequencies $K$ in the Fourier space, we analyze the magnitude of the Fourier coefficients $\|X_w\|$ in the Fourier Space.

The most important frequencies can be identified by examining the magnitude spectrum $\|X_w\|$:

$$\|X_w\| = \sqrt{\Re(X_w)^2 + \Im(X_w)^2} \tag{24}$$

where $\Re(X_w)$ and $\Im(X_w)$ are the real and imaginary parts of the Fourier coefficient $X_W$, respectively.

The frequencies $k$ with the largest magnitudes $\|X_k\|$ are considered the most important, as they contribute the most to the overall structure of the time-series. These frequencies can be identified by finding the local maxima in the magnitude spectrum, also known as the peaks:

$$\text{Important frequencies: } k_i \text{ where } \|X_{k_i}\| > \delta^* \tag{25}$$

where $\delta^*$ is a threshold value calculated as $d$ standard deviations above the mean magnitude in Fourier Space, for each sample. Similarly, it is also possible to define a fixed number of frequencies, so that $\delta^* = \max\{\delta \in \mathbb{R}^+ : |\{X_{k_i} \in \mathbb{C} : \|X_{k_i}\| > \delta\}| \geq k\}$ and there are $k$ stages during the diffusion process.

After identifying the more relevant frequencies, the stages are ordered in ascending amplitudes, such that smaller components are diffused first. The residual is defined as

$$r = z - \sum_{i=0}^{K-1} F_k,$$

where $F$ are the set of frequencies obtained using Eq. equation 22. This means that unlike the traditional method of additive time-series decomposition into seasonality, trend and residual, we combine the trend and residual components into a single component. This is due to the usually simple form, usually monotonic, that the trend component takes. Other works that use timeseries decompositions layers Oreshkin et al. (2020); Fons et al. (2024), have used a low polynomial regressor ($p = 2$) to model the trend.

Using Fourier decomposition, the model can process these components independently. Although we present an implementation of this work using Fourier Analysis to decompose the time-series, any other decomposition, such as wavelet decomposition or more simple decompositions methods can be used.

## A.2    DATASET DETAILS

To benchmark this forward process against the models using the original diffusion process, we utilize four benchmark multivariate real-world datasets commonly found in the literature Shen & Kwok (2023); Alcaraz et al. (2022); Zhou et al. (2021). Unless specified, all data splits and pre-processing was obtained from publicly available repositories to guarantee reproducibility and to facilitate comparisons.

### PTB-XL

The PTB-XL dataset contains 21837 clinical 12-lead electrocardiograms (ECGs), with 1000 timesteps (10s) each, from 18885 patients Wagner et al. (2020). For the forecasting task, the first 900 time-steps where used as conditional data. While the authors understand that forecasting is not a task usually performed for ECG data, we add this dataset for comparison with other works Tashiro et al. (2021); Alcaraz et al. (2022) that used the same dataset.

### ELECTRICITY

It contains univariate electricity usage data (in kWh) gathered from 370 clients every 15 min for over two years. it has missing samples, representing clients who joined during the data gathering process.Trindade (2015). To facilitate training, some authors Alcaraz et al. (2022) have suggested using feature sampling, reducing the channel space to 37, instead of the original 370. In this work, we

Table 3: PTB-XL data set details.

| DETAILS / FORECAST TASK | |
|---|---|
| TRAIN SIZE | 17,441 |
| TEST SIZE | 2,203 |
| TRAINING BATCH | 128 / 32 / 16 |
| SAMPLE LENGTH | 1000 |
| SAMPLE FEATURES | 12 |
| CONDITIONAL VALUES | 800 |
| TARGET VALUES | 200 |
| MAX COMPONENTS | 3 |

expand the dataset by seeing each client has a independent and univariate time-series, also expanding, sample-wise, the training and testing datasets. The test set was undersampled to match the size of the training set, and to reduce inference computational time.

Table 4: Electricity data set details.

| DETAILS / FORECAST TASK | |
|---|---|
| TRAIN SIZE | 302,290 |
| TEST SIZE | 302,290 |
| TRAINING BATCH | 64 |
| SAMPLE LENGTH | 1000 |
| SAMPLE FEATURES | 1 |
| CONDITIONAL VALUES | 76 |
| TARGET VALUES | 24 |
| MAX COMPONENTS | 3 |

MUJOCO

Mujoco (Multi-Joint dynamics with Contact) is a dataset used for physical simulations Rubanova et al. (2019), and it was adapted to time-series by Shan et al. (2023). It has 14 features, each a joint of the physical object being modeled. For the most part, this dataset does not have seasonal features, and many features are mostly constant during the 100 steps of the simulation. It serves to show how our forward diffusion process reduces to the classical formulation when no seasonal features are available.

Table 5: MuJoCo data set details.

| DETAILS / FORECAST TASK | |
|---|---|
| TRAIN SIZE | 8,000 |
| TEST SIZE | 10,000 |
| TRAINING BATCH | 64 |
| SAMPLE LENGTH | 100 |
| SAMPLE FEATURES | 14 |
| CONDITIONAL VALUES | 90 |
| TARGET VALUES | 10 |
| MAX COMPONENTS | 2 |

ETTM1

ETTm1 is a known baseline long-time series dataset for Electricity Transformer Temperature Zhou et al. (2021). The data set contains information from a compilation of 2-year data from two distinct

Chinese counties. We used the ETTm1 version, which cointains time-steps at 15 minutes intervals. The data has a target feature, oil temperature, and six power load covariates. In this work, we used the original preprocessing of data, in 12/4/4 months for train/val/test. The complete information is present in

Table 6: ETTm1 data set details.

| DETAILS / FORECAST TASK | |
|---|---|
| TRAIN SIZE | 33,200 |
| TEST SIZE | 10,000 |
| TRAINING BATCH | 64 |
| SAMPLE LENGTH | 1056 |
| SAMPLE FEATURES | 7 |
| CONDITIONAL VALUES | 864 |
| TARGET VALUES | 192 |
| MAX COMPONENTS | 4 |

### A.3 TEMPERATURE

This dataset contains 32072 daily time series showing the temperature observations and rain forecasts, gathered by the Australian Bureau of Meteorology for 422 weather stations across Australia, between 02/05/2015 and 26/04/2017 Australian Government (2016). Each station observes mean temperature in Celcius averaged over a 24 hour period. This dataset therefore cointains 422 samples with 725 time-steps each. In this paper we use only the target variable mean temperature. The Sashimi model, in its original code, does not support odd length conditional windows, so the length of the conditional time-series is reduced by one.

### A.4 SOLAR-ENERGY

This dataset is related to solar power production in the year of 2006, from 137 PV plants in Alabama State Laboratory (2025). Similarly to the electricity dataset, Each station is treated as an individual sample. leading to 8704 individual time-series with length 192.

Table 7: Temperature data set details.

| DETAILS / FORECAST TASK | |
|---|---|
| TRAIN SIZE | 76 |
| TEST SIZE | 46 |
| TRAINING BATCH | 128 |
| SAMPLE LENGTH | 725 /725 |
| SAMPLE FEATURES | 1 |
| CONDITIONAL VALUES | 628 |
| TARGET VALUES | 96 |
| MAX COMPONENTS | 4 |

### A.5 MODEL HYPERPARAMETERS

In Tables 9 to 12 we indicate the hyperparameters used to train the Models. Following the hyperparameters presented in Alcaraz et al. (2022), we use the same model for all datasets.

Table 8: Solar data set details.

| DETAILS / FORECAST TASK | |
|---|---|
| TRAIN SIZE | 7680 |
| TEST SIZE | 1024 |
| TRAINING BATCH | 128 |
| SAMPLE LENGTH | 192 |
| SAMPLE FEATURES | 1 |
| CONDITIONAL VALUES | 24 |
| TARGET VALUES | 24 |
| MAX COMPONENTS | 3 |

Table 9: SSSD hyperparameters.

| HYPERPARAMETER | VALUE |
|---|---|
| RESIDUAL LAYERS | 36 |
| RESIDUAL CHANNELS | 256 |
| SKIP CHANNELS | 256 |
| DIFFUSION EMBEDDING DIM. 1 | 128 |
| FULLY CONNECTED EMBEDDING | 512 |
| BIDIRECTIONAL S4 SIZE | 64 |
| SCHEDULE | LINEAR |
| DIFFUSION STEPS | $T$ 200 |
| $B_0$ | 0.001 |
| $B_T$ | 0.02 |
| OPTIMIZER | ADAM |
| LOSS FUNCTION | MSE |
| LEARNING RATE | $2 \times 10^{-4}$ |

Table 10: DiffWave hyperparameters.

| HYPERPARAMETER | VALUE |
|---|---|
| RESIDUAL LAYERS | 36 |
| RESIDUAL CHANNELS | 256 |
| SKIP CHANNELS | 256 |
| DIFFUSION EMBEDDING DIM. 1 | 128 |
| FULLY CONNECTED EMBEDDING | 512 |
| SCHEDULE | LINEAR |
| DIFFUSION STEPS $T$ | 200 |
| $B_0$ | 0.001 |
| $B_T$ | 0.02 |
| OPTIMIZER | ADAM |
| LOSS FUNCTION | MSE |
| LEARNING RATE | $2 \times 10^{-4}$ |

## A.6 DIFFUSION STEP EMBEDDING

It is necessary to incorporate the diffusion step and stage $t, k$ as part of the input, since the model needs to produce different $\epsilon'_\theta(z_k^t, t, k)$ for various $t, k$. Compared with other models, we have one more term $(k)$ that represents the stage; however, the incorporation is straightforward, for a maximum value $K_{max}$ that defines the maximum number of components, each diffusion step is equal to $t^* = t + \tau(k-1)$.

We use a 128-dimensional encoding vector for each $t$ based on Vaswani et al. (2017), and as already shown in Kong et al. (2021):

Table 11: Sashimi hyperparameters.

| HYPERPARAMETER | VALUE |
|---|---|
| RESIDUAL LAYERS | 6 |
| POOLING FACTOR | [2,2] |
| FEATURE EXPANSION | 2 |
| DIFFUSION EMBEDDING DIM. 1 | 128 |
| FULLY CONNECTED EMBEDDING | 512 |
| SCHEDULE | LINEAR |
| DIFFUSION STEPS | $T$ 200 |
| $B_0$ | 0.001 |
| $B_T$ | 0.02 |
| OPTIMIZER | ADAM |
| LOSS FUNCTION | MSE |
| LEARNING RATE | $2 \times 10{-4}$ |

Table 12: CSDI hyperparameters.

| HYPERPARAMETER | VALUE |
|---|---|
| RESIDUAL LAYERS | 4 |
| RESIDUAL CHANNELS | 64 |
| DIFFUSION EMBEDDING DIM. | 128 |
| FULLY CONNECTED EMBEDDING | 128 |
| SCHEDULE | LINEAR |
| DIFFUSION STEPS $T$ | 200 |
| $B_0$ | 0.001 |
| $B_T$ | 0.02 |
| FEATURE EMBEDDING DIM. | 128 |
| TIME EMBEDDING DIM. | 16 |
| OPTIMIZER | ADAM |
| LOSS FUNCTION | MSE |
| LEARNING RATE | $1 \times 10^{-3}$ |
| WEIGHT DECAY | $10^{-6}$ |

$$t_{\text{embedding}} = \left[ \sin\left(10^{0 \times \frac{4}{63}} t\right), \ldots, \sin\left(10^{\frac{63 \times 4}{63}} t\right), \cos\left(10^{0 \times \frac{4}{63}} t\right), \ldots, \cos\left(10^{\frac{63 \times 4}{63}} t\right) \right] \quad (26)$$

## A.7 METRICS

### A.7.1 MSE

The mean squared error (MSE) is a standard point-forecast accuracy metric. Given predicted values $\hat{y}_{t,c}$ and ground-truth values $y_{t,c}$ for $T$ time steps and $C$ features, it is defined as

$$\text{MSE} = \frac{1}{C} \sum_{c=1}^{C} \frac{1}{T} \sum_{t=1}^{T} (y_{t,c} - \hat{y}_{t,c})^2. \quad (27)$$

## A.8 MAE

Given predicted values $\hat{y}_{t,c}$ and ground-truth values $y_{t,c}$ for $T$ time steps and $C$ features, it is defined as

$$\text{MAE} = \frac{1}{C} \sum_{c=1}^{C} \frac{1}{T} \sum_{t=1}^{T} |y_{t,c} - \hat{y}_{t,c}|. \quad (28)$$

### A.8.1 CRPS

We describe the definition and computation of the CRPS metric. The continuous ranked probability score (CRPS) Matheson & Winkler (1976) measures the compatibility of an estimated probability distribution $F$ with an observation $\hat{y}$, and can be defined as the integral of the quantile loss $\Lambda_\alpha(q, z) := (\alpha - \mathbf{1}_{z<q})(z - q)$ for all quantile levels $\alpha \in [0, 1]$:

$$\mathrm{CRPS}(F^{-1}, \hat{y}) = \int_0^1 2\Lambda_\alpha(F^{-1}(\alpha), \hat{y}) \, d\alpha. \tag{29}$$

where $\mathbf{1}$ is the indicator function. Following Tashiro et al. (2021), we generated 100 independent samples to approximate the distribution $F$ over each forecast time step and feature. We computed quantile losses for discretized quantile levels with 0.05 ticks in the set [0.05,0.95]. Namely, we approximated CRPS with

$$\mathrm{CRPS}(F^{-1}, \hat{y}) \approx \frac{2}{19} \sum_{i=1}^{19} 2\Lambda_{i \cdot 0.05}(F^{-1}(i \cdot 0.05), \hat{y}). \tag{30}$$

averaging the result across time-steps and features. This metric, due to computational constrains, is only used to evaluate the ablation studies.

# B  DERIVATION OF EQUATIONS EQUATION 19 AND EQUATION 21

## B.1  PROPOSITION 1

For any given $k$, the terms $\sum_{n>k}^{K} f_0^n$ and $\sum_{n=0}^{k} d_n$ are kept constant for all $t \in [0, T[$ and thus, using Eq. equation 15 in equation $q(f_{t-1}^k | f_t^k, f_0^k) \propto q(f_t^k | f_{t-1}^k) q(f_{t-1}^k | f_0^k)$ we can obtain the distribution of the reverse diffusion process for each component.

$$
\begin{aligned}
q(f_{t-1}^k | f_t^k, f_0^k) &= \frac{q(f_t^k | f_{t-1}^k) q(f_{t-1}^k | f_0^k)}{q(f_t^k | x_0)} \\
&\propto \mathcal{N}(f_t^k; \sqrt{\alpha_t} f_{t-1}^k, d_k(1-\alpha_t)\mathbf{I}) \mathcal{N}(f_{t-1}^k; \sqrt{\bar{\alpha}_{t-1}} f_0^k, d_k(1-\bar{\alpha}_{t-1})\mathbf{I}) \\
&\propto \exp\left\{ -\left[ \frac{(f_t^k - \sqrt{\alpha_t} f_{t-1}^k)^2}{2 d_k(1-\alpha_t)} + \frac{(f_{t-1}^k - \sqrt{\bar{\alpha}_{t-1}} f_0^k)^2}{2 d_k(1-\bar{\alpha}_{t-1})} \right] \right\} \\
&= \exp\left\{ -\frac{1}{2 d_k} \left[ \frac{(f_t^k - \sqrt{\alpha_t} f_{t-1}^k)^2}{1-\alpha_t} + \frac{(f_{t-1}^k - \sqrt{\bar{\alpha}_{t-1}} f_0^k)^2}{1-\bar{\alpha}_{t-1}} \right] \right\} \\
&= \exp\left\{ -\frac{1}{2 d_k} \left[ \frac{-(2\sqrt{\alpha_t} f_t^k f_{t-1}^k)}{1-\alpha_t} + \frac{\alpha_t(f_{t-1}^k)^2}{1-\alpha_t} + \frac{(f_{t-1}^k)^2}{1-\bar{\alpha}_{t-1}} - \frac{2\sqrt{\bar{\alpha}_{t-1}} f_{t-1}^k f_0^k}{1-\bar{\alpha}_{t-1}} + C(f_t^k, f_0^k) \right] \right\} \\
&= \exp\left\{ -\frac{1}{2 d_k} \left[ \left( \frac{\alpha_t}{1-\alpha_t} + \frac{1}{1-\bar{\alpha}_{t-1}} \right)(f_{t-1}^k)^2 - 2\left( \frac{\sqrt{\alpha_t} f_t^k}{1-\alpha_t} + \frac{\sqrt{\bar{\alpha}_{t-1}} f_0^k}{1-\bar{\alpha}_{t-1}} \right) f_{t-1}^k \right] \right\} \\
&= \exp\left\{ -\frac{1}{2 d_k} \left[ \frac{\alpha_t(1-\bar{\alpha}_{t-1}) + 1 - \alpha_t}{(1-\alpha_t)(1-\bar{\alpha}_{t-1})}(f_{t-1}^k)^2 - 2\left( \frac{\sqrt{\alpha_t} f_t^k}{1-\alpha_t} + \frac{\sqrt{\bar{\alpha}_{t-1}} f_0^k}{1-\bar{\alpha}_{t-1}} \right) f_{t-1}^k \right] \right\} \\
&= \exp\left\{ -\frac{1}{2 d_k} \left[ \frac{1-\bar{\alpha}_t}{(1-\alpha_t)(1-\bar{\alpha}_{t-1})}(f_{t-1}^k)^2 - 2\left( \frac{\sqrt{\alpha_t} f_t^k}{1-\alpha_t} + \frac{\sqrt{\bar{\alpha}_{t-1}} f_0^k}{1-\bar{\alpha}_{t-1}} \right) f_{t-1}^k \right] \right\} \\
&= \exp\left\{ -\frac{1}{2} \frac{1-\bar{\alpha}_t}{d_k(1-\alpha_t)(1-\bar{\alpha}_{t-1})} \left[ (f_{t-1}^k)^2 - 2 \frac{\left( \frac{\sqrt{\alpha_t} f_t^k}{1-\alpha_t} + \frac{\sqrt{\bar{\alpha}_{t-1}} f_0^k}{1-\bar{\alpha}_{t-1}} \right)}{\frac{1-\bar{\alpha}_t}{(1-\alpha_t)(1-\bar{\alpha}_{t-1})}} f_{t-1}^k \right] \right\} \\
&= \exp\left\{ -\frac{1}{2} \frac{1-\bar{\alpha}_t}{d_k(1-\alpha_t)(1-\bar{\alpha}_{t-1})} \left[ (f_{t-1}^k)^2 - 2 \frac{\left( \frac{\sqrt{\alpha_t} f_t^k}{1-\alpha_t} + \frac{\sqrt{\bar{\alpha}_{t-1}} f_0^k}{1-\bar{\alpha}_{t-1}} \right)(1-\alpha_t)(1-\bar{\alpha}_{t-1})}{1-\bar{\alpha}_t} f_{t-1}^k \right] \right\} \\
&= \exp\left\{ -\frac{1}{2} \frac{1}{\frac{d_k(1-\alpha_t)(1-\bar{\alpha}_{t-1})}{1-\bar{\alpha}_t}} \left[ (f_{t-1}^k)^2 - 2 \frac{\sqrt{\alpha_t} f_t^k(1-\bar{\alpha}_{t-1}) + \sqrt{\bar{\alpha}_{t-1}} f_0^k(1-\alpha_t)}{1-\bar{\alpha}_t} f_{t-1}^k \right] \right\} \\
&\propto \mathcal{N}\left( f_t^k; \mu_q = \frac{\sqrt{\alpha_t} f_t^k(1-\bar{\alpha}_{t-1}) + \sqrt{\bar{\alpha}_{t-1}} f_0^k(1-\alpha_t)}{1-\bar{\alpha}_t}, \sigma_q = \frac{d_k(1-\alpha_t)(1-\bar{\alpha}_{t-1})}{1-\bar{\alpha}_t} \mathcal{I} \right)
\end{aligned}
\tag{31}
$$

where $C(f_t^0, f_t^k)$ are the terms that complete the square, and are constant in respect to $f_{t-1}^k$. Effectively, this formulation is similar to the original, added the $d_k$ factor to $\sigma_q = \frac{d_k(1-\alpha_t)(1-\bar{\alpha}_{t-1})}{1-\bar{\alpha}_t}$ To complete the diffusion process, we know that each reverse diffusion $q$ is normally distributed, and that at each $t = 0$, $z_0^k \sim \mathcal{N}(\sum_{n>k}^{K} f_0^n, \sum_{n=0}^{k} d_n)$.

Now using the reparametrization trick already defined in Eq. equation 19 for any $k$, we have that:

$$f_0^k = \frac{z_t^k - \sqrt{-d_k \bar{\alpha}_t + \sum_{n=0}^k d_n} \epsilon - \sum_{n>k}^K f_0^n}{\sqrt{\bar{\alpha}_t}}$$

$$f_0^k = \frac{z_t^k - \epsilon' - \sum_{n>k}^K f_0^n}{\sqrt{\bar{\alpha}_t}}$$

$$f_0^k = \frac{f_t^k - \epsilon'}{\sqrt{\bar{\alpha}_t}},$$

and replacing that in Eq. equation 31, we obtain:

$$\mu_q = \frac{\sqrt{\alpha_t}(1 - \bar{\alpha}_{t-1})f_t^k + \sqrt{\bar{\alpha}_{t-1}}(1 - \alpha_t)f_0^k}{1 - \bar{\alpha}_t}$$

$$= \frac{\sqrt{\alpha_t}(1 - \bar{\alpha}_{t-1})f^k t + \sqrt{\bar{\alpha}_{t-1}}(1 - \alpha_t)\frac{f_t^k - \epsilon'}{\sqrt{\bar{\alpha}_t}}}{1 - \bar{\alpha}_t}$$

$$= \frac{\sqrt{\alpha_t}(1 - \bar{\alpha}_{t-1})f_t^k + (1 - \alpha_t)\frac{f_t^k - \epsilon'}{\sqrt{\alpha_t}}}{1 - \bar{\alpha}_t}$$

$$= \frac{\sqrt{\alpha_t}(1 - \bar{\alpha}_{t-1})f_t^k}{1 - \bar{\alpha}_t} + \frac{(1 - \alpha_t)f_t^k}{(1 - \bar{\alpha}_t)\sqrt{\alpha_t}} - \frac{(1 - \alpha_t)\epsilon'}{(1 - \bar{\alpha}_t)\sqrt{\alpha_t}}$$

$$= \left(\frac{\sqrt{\alpha_t}(1 - \bar{\alpha}_{t-1})}{1 - \bar{\alpha}_t} + \frac{1 - \alpha_t}{(1 - \bar{\alpha}_t)\sqrt{\alpha_t}}\right)f_t^k - \frac{(1 - \alpha_t)\epsilon'}{(1 - \bar{\alpha}_t)\sqrt{\alpha_t}}$$

$$= \left(\frac{\alpha_t(1 - \bar{\alpha}_{t-1})}{(1 - \bar{\alpha}_t)\sqrt{\alpha_t}} + \frac{1 - \alpha_t}{(1 - \bar{\alpha}_t)\sqrt{\alpha_t}}\right)f_t^k - \frac{(1 - \alpha_t)\epsilon'}{(1 - \bar{\alpha}_t)\sqrt{\alpha_t}}$$

$$= \frac{(\alpha_t - \alpha_t + 1 - \bar{\alpha}_t)}{(1 - \bar{\alpha}_t)\sqrt{\alpha_t}}f_t^k - \frac{(1 - \alpha_t)\epsilon'}{(1 - \bar{\alpha}_t)\sqrt{\alpha_t}}$$

$$= \frac{1}{\sqrt{\alpha_t}}f_t^k - \frac{(1 - \alpha_t)}{(1 - \bar{\alpha}_t)\sqrt{\alpha_t}}\epsilon' \tag{32}$$

Since $\mu(z_t^k, \Theta, t, k)$ follows the formula of $\mu_q$ we set our denoising formula as:

$$\mu(\mathbf{z}_t^k, \Theta, t, k) = \frac{1}{\sqrt{\alpha_t}}\left(\mathbf{f}_t^k - \frac{1 - \alpha_t}{1 - \bar{\alpha}_t}\boldsymbol{\epsilon}'_\theta(\mathbf{z}_t^k, t, k)\right) + \sum_{n>k}^K \mathbf{f}_0^n \tag{33}$$

with noise:

$$\sigma(\alpha, \hat{d}_k) = \frac{\hat{d}_k(1 - \alpha_t)(1 - \bar{\alpha}_{t-1})}{1 - \bar{\alpha}_t} \tag{34}$$

## C  PROPOSITION ON THE SNR OF THE STAGED DIFFUSION PROCESS

In this section we present the SNR of the baseline diffusion process, and of our staged diffusion process.

### C.1  SETUP AND DEFINITIONS

Let the lossless decomposition be $x_0 = \sum_{k=1}^K f_0^k$ with component energies $d_k = \mathrm{E}[(f_0^k)^2]$ and where components are ordered by amplitude $d_1 < d_2 < d_3 < d_K$.

The staged forward diffusion process diffuses component 1 first, then 2, all the way to $K$. Within each stage the forward update is

$f_t^k = \sqrt{\bar{\alpha}_t^*} + \sqrt{d_k(1 - \bar{\alpha}_t^*)}\epsilon, \quad \epsilon \sim N(0,1)$ At stage $k$ and step $t$ the observable state is

$$z_t^k = f_t^k + \underbrace{\sum_{n>k} f_0^n}_{\text{not yet diffused (clean)}} + \underbrace{(\sum_{m<k} d_m)\epsilon}_{\text{Gaussian noise from past stages}} \quad .$$

For the usual diffusion process, the SNR is:

$$\text{SNR}^{(t)} := \frac{\mathbb{E}\left[\|\sqrt{\bar{\alpha}_t}\, x_0\|^2\right]}{\mathbb{E}\left[\|\sqrt{1 - \bar{\alpha}_t}\, \epsilon\|^2\right]} = \frac{\bar{\alpha}_t \text{Var}(x_0)}{(1 - \bar{\alpha}_t)\text{Var}(\epsilon)} = \frac{\bar{\alpha}_t}{1 - \bar{\alpha}_t}. \tag{35}$$

assuming that $x_0$ is normalized with unit variance.

## C.2 PROPERTY 1.

For each component of our staged diffusion process:

$$\text{SNR}_k^{(t)} := \frac{\mathbb{E}\,|\sqrt{\bar{\alpha}_t^*}\, f_0^k|^2}{\mathbb{E}\,|\sqrt{d_k(1 - \bar{\alpha}_t^*)}\, \varepsilon|^2} = \frac{\bar{\alpha}^*_t d_k}{d_k(1 - \bar{\alpha}^*_t)}.$$

The $\text{SNR}_k^{(t)}$ is strictly decreasing in $t$.

$$\text{SNR}_k^{(t)} = \frac{\bar{\alpha}_t^*}{1 - \bar{\alpha}_t^*},$$

note that the SNR equation is quite similar to 35, with a different sequence of $\{\bar{\alpha}^*\}_{t=1}^{\tau}$, therefore we know that within each stage the SNR decays smoothly in a known way..

## C.3 PROPERTY 2

At any global step, with component index k and diffusion step t, the total SNR for the observable variable $z_t^k$ is:

$$TSNR = \frac{\mathbb{E}\left[|f_t^k + \sum_{n>k} f_0^n|^2\right]}{\mathbb{E}\left[\left|\sqrt{d_k(1 - \bar{\alpha}_t^*)} + \sum_{m<k} d_m\, \epsilon\right|^2\right]} \tag{36}$$

$$TSNR = \frac{d_k\bar{\alpha}_t^* + \sum_{n>k} d_n}{-d_k\bar{\alpha}_t^* + \sum_{m\leq k} d_m}. \tag{37}$$

and the TSNR at the end of each stage is simply:

$$TSNR = \frac{\sum_{n>k} d_n}{\sum_{m\leq k} d_m} \quad \text{for } t = \tau. \tag{38}$$

given that $\alpha_t$ is designed to converge to 0. Since $d_k$ is strictly increasing and monotone, we first can guarantee that TSNR >0, it is monotone, and that it tends to 0 when $t \to \tau$ and $k = K$

Also we can define an upper bound on the TSNR at the end of each stage is $d_1 = d_2 = ... = d_K = d$.

and that the TSNR defined by that upper bound is:

$$TSNR^+(k) = \frac{K - k}{k} \tag{39}$$

for a TSNR that is only defined at the end of each stage, and knowing that $d = 1/K$.

## D  ABLATION STUDIES

### SYNTHETIC DATASET

This synthetic dataset contains 10000 samples of a signal composed with three different frequencies, with 1000 time-steps each. The goal of this synthetic dataset is to evaluate how the forward diffusion process decomposes the signal, and how the different signals are recuperated during the sampling process.

---

**Algorithm 3** Synthetic Dataset Generation

---

1: **Input:** $N \in \mathbb{N}$ (number of samples)
2: **Output:** $\{y^i\}_{i=1}^N$ (synthetic time series)
3: $f_s \leftarrow 1000$ {Sampling frequency}
4: $t \leftarrow \{0, \frac{1}{f_s}, \frac{2}{f_s}, \ldots, 1\}$ {Time grid}
5:
6: **for** $i = 1$ to $N$ **do**
7:     $\gamma_{f1}^{(i)} \sim \text{Gamma}(\alpha = 3, \beta = 5)$
8:     $\gamma_{f2}^{(i)} \sim \text{Gamma}(\alpha = 4, \beta = 5)$
9:     $\gamma_{f3}^{(i)} \sim \text{Gamma}(\alpha = 5, \beta = 5)$
10:     $\left(\gamma_{f1}^{(i)}, \gamma_{f2}^{(i)}, \gamma_{f3}^{(i)}\right) \leftarrow \text{sort}\left(\gamma_{f1}^{(i)}, \gamma_{f2}^{(i)}, \gamma_{f3}^{(i)}\right)$
11: **end for**
12:
13: **for** $i = 1$ to $N$ **do**
14:     $a_2^{(i)} \sim \text{Uniform}(0.1, 1)$
15:     $a_3^{(i)} \sim \text{Uniform}(0.1, 1)$
16: **end for**
17: **for** $i = 1$ to $N$ **do**
18:     $y_1^i(t) \leftarrow \sin(2\pi \gamma_{f1}^{(i)} t)$
19:     $y_2^i(t) \leftarrow a_2^{(i)} \sin(2\pi \gamma_{f2}^{(i)} t)$
20:     $y_3^i(t) \leftarrow a_3^{(i)} \sin(2\pi \gamma_{f3}^{(i)} t)$
21: **end for**
22: $y = y_1 + y_2 + y_3$

---

Table 13: Synthetic data set details.

| DETAILS / FORECAST TASK | |
| --- | --- |
| TRAIN SIZE | 9000 |
| TEST SIZE | 1000 |
| TRAINING BATCH | 264 |
| SAMPLE LENGTH | 1000 |
| SAMPLE FEATURES | 1 |
| CONDITIONAL VALUES | 700 |
| TARGET VALUES | 300 |
| MAX COMPONENTS | 3 |

### D.1  COMPONENTS ABLATION

In this section we evaluate the component number. This hyperparameter can be either fixed or variable. In the fixed setting, the number of components is fixed and given as a hyperparameter, in the variable setting, the hyperparameter is only an upper bound. Knowing that the ground truth value of components for this dataset is 3, we evaluate if the diffusion model with the appropriate number of components obtains better results. All models were trained using the same architecture and the same number of training steps, with the best model in the validation dataset used for testing.

Table 14: Forecasting results on synth1 with varying number of components. The ground truth value of components is n=3.

| COMPONENTS ($n$) | MSE↓ | MAPE↓ | MAE↓ | CRPS↓ |
|---|---|---|---|---|
| 5 | 0.120 | 2.809 | 0.233 | 0.0636 |
| 4 | 0.189 | 4.147 | 0.307 | 0.0826 |
| **3** (TRUE VALUE) | **0.0936** | **1.794** | 0.197 | 0.0510 |
| 2 | 0.157 | 3.57 | 0.271 | 0.0712 |
| 1 | 0.0978 | 2.056 | **0.185** | **0.0431** |
| 0 (BASELINE) | 0.0948 | 2.620 | 0.204 | 0.0550 |

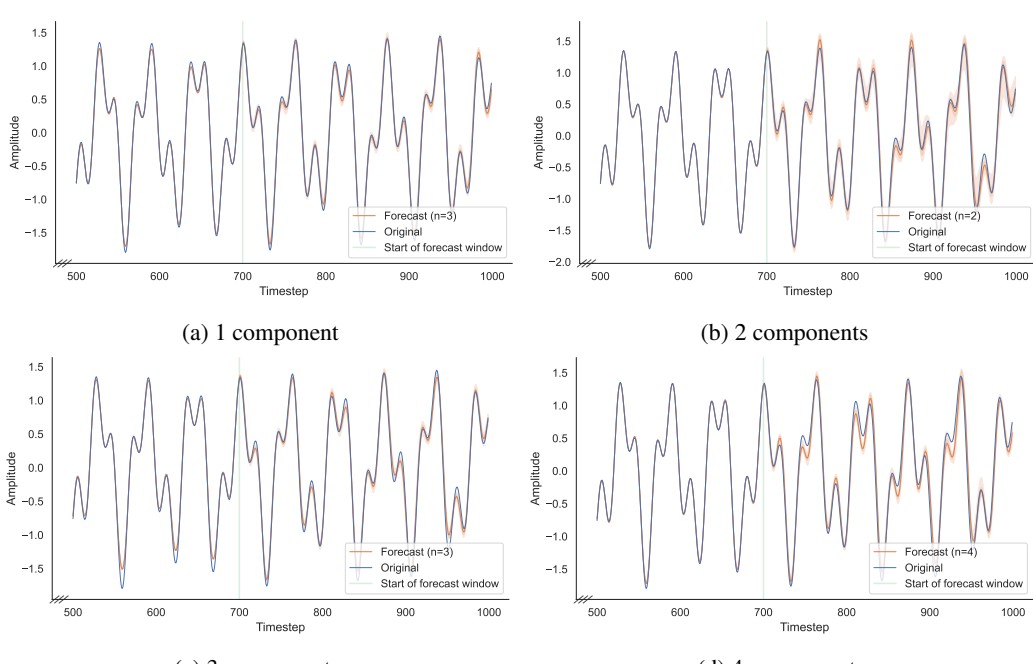

(a) 1 component

(b) 2 components

(c) 3 components

(d) 4 components

Figure 3: forecasting comparison for a 300 steps window length, of choosing a different number of components.

By evaluating the forecast quality, when choosing different number of components, while keeping all other parameters the same. We can first observe that using the appropriate number of components yields better result. Of the models with the improper number of components, the model with just one component had the best results, possibly because it divides the diffusion process into two

The baseline, which here represents the usual diffusion process, provides a comparable forecast with the correct number of components, having just slight worst results, especially in MAPE.

From this we can conclude that the components based diffusion has a significant effect on the diffusion process.

## D.2 SCHEDULER PARAMETERS

This ablation study analyzes how the scheduler is affected. For a fixed T, as the number of components increase, the number of diffusion steps per component decrease. This leads to a less smooth, not continuous forward diffusion process, with gaps when transitioning between components. Furthermore the last component is, by definition, the one with smaller amplitude, and therefore, if more components than necessary are added, the final components of the reverse diffusion have a very limited impact. In Fig. 4 we can observe this effect using a toy example, with the average noising process of

1000 training examples, as we add components and fix the scheduler, using ($\beta_0 = 0.01, \beta_T = 0.05$). The more components are added, the less smooth is the forward diffusion process, and also, for $n \geq 3$, the forward diffusion process does not reach the usual diffusion standard deviation of 1.

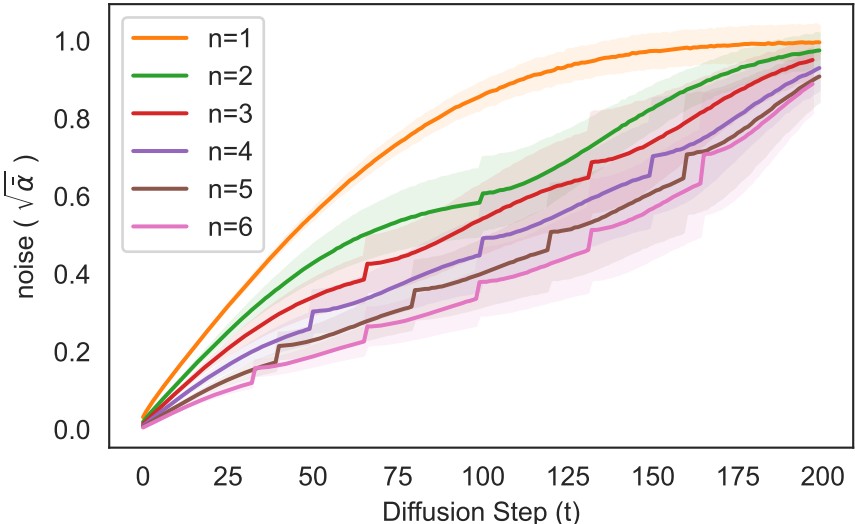

Figure 4: Evolution of the noise scheduler with different number of components. As the number of components increase, the smoothness of the diffusion process decreases, for fixed scheduler parameters.

To solve this, we identify the parameter space that is able to satisfy our requirements of smoothness, and guarantee that the final step of the diffusion process has $z_T^K \sim N(0, 1)$.

Given a linear noise schedule in a diffusion process, we want it to satisfy a global bound on cumulative noise $\bar{\alpha}_\tau = \prod_{t=0}^{\tau} \alpha$. Let $\tau$ denote the total number of diffusion steps for a given component, and let $\beta_0$ and $\beta_\tau$ be the starting and ending noise parameters, respectively. A linear scheduler can be trivially defined as:

$$\beta_t = \beta_0 + \frac{t}{\tau}(\beta_\tau - \beta_0), \quad \text{for } t = 0, 1, \ldots, \tau.$$

Our goal is to enforce the constraint that $\bar{\alpha}_\tau$ is bellow a threshold $q$:

$$\prod_{t=0}^{\tau}(1 - \beta_t) \leq q, \tag{40}$$

where typically $q \in ]0, 0.1]$, and considering the constraints $b_\tau > b_0$ and $b_\tau, b_0 > 0$.

The solution can be solved numerically for $\tau, \beta_\tau$ and $\beta_0$ as seen on Fig.5. Note that these values are independent of the data. Since, for the usual $T = 200$, $\tau$ depends on the number of components, and choosing a fixed $\beta_0 = 0.001$ we can approximate the minimum value of $\beta_\tau$ that follows the constraint.

By choosing appropriate scheduler parameters, a smoother, more continuous growth of the noise added to the sample is obtained, as can clearly be see in Fig. 6. Also, this guarantees that $\sqrt{\bar{\alpha}_t}$ converges to 1 at the end of the diffusion process.

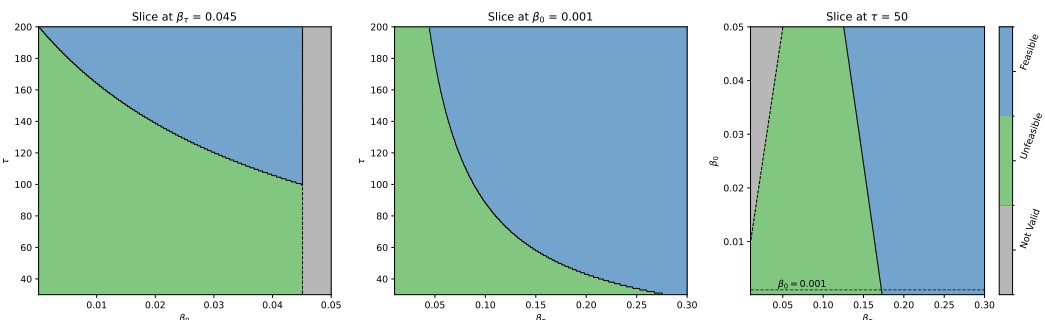

Figure 5: Exploration of the feasibility region for $T \in [30, 200], \beta_\tau \in [0.01, 0.3], \beta_0 \in [0.0001, 0.05]$. The gray area indicates where $b_\tau > b_0$, with the green area indicating the feasibility space.

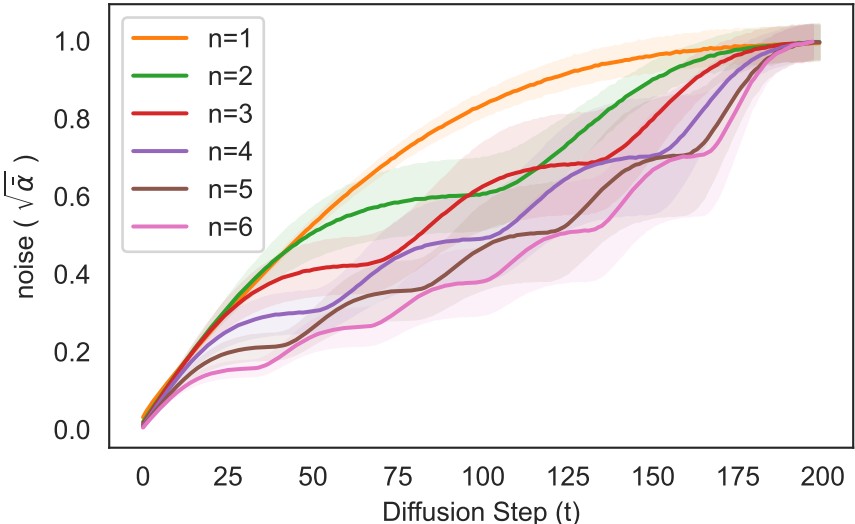

Figure 6: Evolution of the noise scheduler, with tuned $\beta_\tau$. By tuning $\beta_\tau$, the noise scheduler is able to reach a value close to zero when $t = \tau$, regardless of the number of components, or the average amplitude of each component.

## E  COMPUTATIONAL COST

We analyze both the theoretical and empirical computational cost of our decomposition-based diffusion model.

### E.1  THEORETICAL COST

Our decomposition function requires performing a Fast Fourier Transform (FFT) over the sample batch, selecting the top-$k$ frequencies, and then applying an inverse FFT (IFFT) to reconstruct the chosen components. The sorting step is negligible, since the number of selected components $k$ is much smaller than the the number of time steps $N$.

Overall, the computational complexity of our method is:

$$\mathcal{O}(2N \log N). \tag{41}$$

In practice, the additional cost is primarily observed during training, where the decomposition is performed for each batch. During inference, the decomposition is executed only once every $T$ diffusion steps, so the overhead remains very limited.

### E.2 EMPIRICAL COST

We measure the practical overhead in terms of average increase in training time per epoch across multiple datasets and models. The results are summarized in Table 15. We observe an average overhead of approximately 7.5% per epoch, which is modest given the accuracy improvements.

Table 15: Increase in training time per epoch due to FFT-based decomposition.

| Model | Electricity | ETTm1 | Mujoco | PTB-XL | Synth1 | Temperature |
|-------|-------------|-------|--------|--------|--------|-------------|
| FFT-DM | 11.79% | 1.19% | 11.55% | 4.19% | 12.10% | 4.25% |

### E.3 ALGORITHMIC

We summarize the decomposition procedure in Algorithm 4 , for simplicity for just one channel, since the extension is trivial.

---

**Algorithm 4** FFT-based Decomposition

---

1: Batch of time-series $X = \{x_0\}_{i=1}^B$,
2: $\tilde{F} \leftarrow \mathbf{0}_{B \times k \times N}$
3: $F \leftarrow \text{FFT}(X)$
4: $I \leftarrow$ indices of top-$k$ frequencies in each row of $|F|$
5: $\tilde{F} \leftarrow F[:, I]$ {keep only top-$k$ frequencies per sequence, in separate components}
6: $D \leftarrow \text{IFFT}(\tilde{F})$
7:
8: **return** Sort($D$)

---

For completeness, we also propose a simple method to select the number of components using the Fourier domain. Given a dataset, we compute the Fourier transform of each sample and examine the resulting magnitude spectrum. To account for noise and the typical spectral decay at higher frequencies, we focus on local maxima—frequencies whose amplitude exceeds that of their neighbors—rather than absolute amplitude values. By applying a second discrete derivative to the magnitude spectrum, we can identify inflection points, which correspond to these local maxima. The number of detected peaks then provides a principled estimate for the number of relevant frequency components in the data.

## F  RELATED WORK

**c) Neural Networks for Time-series**    Deep learning-based time-series forecasting has evolved through various architectures. Early approaches leveraged recurrent models such as RNNs Tealab (2018) and LSTMs Hochreiter & Schmidhuber (1997). More recently, Transformer-based models, including Autoformer Wu et al. (2021) and Informer Zhou et al. (2021), have achieved state-of-the-art results, with Autoformer replacing self-attention with auto-correlation and Informer utilizing sparse attention for efficiency. Despite their success, these models still struggle with long-range dependencies, particularly in time-series data with strong periodicity, where attention mechanisms fail to capture recurring patterns beyond the lookback window.

**d) Time Series Decomposition**    Time series decomposition methods aim to separate signals into interpretable components such as trend, seasonality, and residual. Classical approaches include moving-average and trend–seasonality decomposition, which rely on rolling and periodic averaging Hyndman & Athanasopoulos (2021). STL decomposition extends these ideas by applying LOESS smoothing, allowing the seasonal component to vary over time Cleveland et al. (1990). A second group of Spectral techniques, such as Singular Spectrum Analysis Broomhead & King (1986), Wavelet Transforms Mallat (1989) and Fourier-based decomposition Cooley & Tukey (1965), which was use used in this work, are designed to capture structure across time and frequency..

# G FORECAST FIGURES

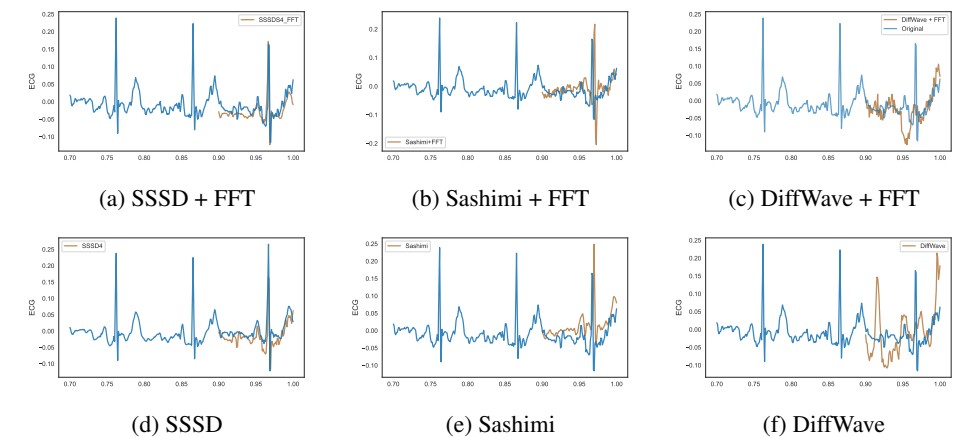

(a) SSSD + FFT  (b) Sashimi + FFT  (c) DiffWave + FFT

(d) SSSD  (e) Sashimi  (f) DiffWave

Figure 7: One generation for a random sample of the PTB-XL dataset, for three different models with and without FFT decomposition.