# OpenReview forum: "FFT-DM: A Decomposable Forward Process in Diffusion Models for Time-Series Forecasting"
_ICLR.cc/2026/Conference — ICLR 2026 Conference Withdrawn Submission_

### Official Review · Reviewer_sxCd · 2025-10-26

**Soundness:** 2
**Presentation:** 2
**Contribution:** 2
**Rating:** 2
**Confidence:** 5

**Summary:**

The paper introduces FFT-DM, a fourier decomposition-based forward diffusion process that decomposes time series into K Fourier components and applies staged noise injection in ascending order of amplitude. The mathematical framework provides closed-form expressions for the forward process (equation 17) and derives reverse diffusion equations with SNR-scaled variance terms. The method modifies standard diffusion training by sampling component indices k alongside timesteps t, requiring noise models to output scaled predictions. Two algorithms specify the complete training and inference procedures, including component reconstruction during the reverse process.

Experiments evaluate four diffusion architectures (DiffWave, SSSD, CSDI, Sashimi) across six datasets spanning electricity demand, temperature, ECG signals, physical simulations, and solar power. Results demonstrate MSE/MAE improvements on datasets with strong periodicity, particularly Electricity and PTB-XL. The paper includes computational overhead analysis showing approximately 7\% training time increase across datasets. Ablation studies on synthetic multi-frequency signals examine the impact of component numbers (K=1 to 5), revealing optimal performance at the true component count of K=3.

Theoretically the authors have attempted to include a formal SNR analysis to prove that component-wise diffusion maintains higher signal-to-noise ratios for dominant frequencies throughout the diffusion trajectory. The appendix provides hyperparameters, specifically the scheduler hyperparameter range showing feasible regions for noise parameters that ensure smooth diffusion convergence. Complete implementation details include Fourier decomposition algorithms, frequency identification methods using magnitude spectrum peaks, dataset preprocessing specifications, model hyperparameters, and visualization of forecast quality on PTB-XL samples comparing all model variants.

**Strengths:**

The major strength is that the authors have thought about the idea to use FFT decomposition and diffusion on each component. It is a good start. Another strength is that the proposed method only modifies the forward/reverse process, not network architecture—demonstrated across 4 different diffusion backbones

**Weaknesses:**

1. The first major weakness is methodological. Real-world time series often exhibit cross-scale interactions, for example, amplitude modulation where a low-frequency trend affects high-frequency oscillations, or frequency-dependent dampening. Why are cross-frequency effects not included? What is the theoretical justification of why diffusion is performed component wise? Shouldn't conditional diffusion be performed? In fact, empirical justification would also have been fine, but empirical results do not show consistent improvement over the limited baselines considered.

2. Compared baselines are insufficient. More non-diffusion models ( PatchTST, TimesNet, DLinear, N-BEATS etc) and other more powerful diffusion models (MrDiff, CNDiff) are missing. Atleast non-diffusion models using fourier decomposition such as TimeMixer (mentioned and cited) should have been used. Improvement over frequency aware non-diffusion baselines would give more confidence in the paper's experimental results.

3. The considered datasets are inadequate to show performance. PTB-XL is not a good choice (the paper has acknowledged this, good move by the authors. But acknowledgment does not make it ok!) ECG has very regular periodic structure that is artificially favorable for the method used. It is better to replace and/or add genuine long-horizon forecasting benchmarks (e.g., Traffic, Weather from recent papers)

4. Algorithm 2 requires estimating SNR values  at inference from conditioning timesteps. Distributional shift is very common in practice and may explain some negative results. Further no ablation is performed on the SNR estimation method. It is unclear how the proposed method is robust to non-stationary data.

5. The paper has limited scope for application. Many real forecasting problems lack clear periodicity (e.g., financial markets, irregular events). The paper doesn't quantify how much periodicity is needed for benefits. For example, no metric is proposed to pre-assess dataset suitability. This limits practical adoption without clear applicability criteria.

6. Paper claims the framework "generalizes to any additive decomposition method" (page 5), but there is no empirical evidence to support the methods mentioned in Appendix A.1 (wavelet or moving-average decomposition) is provided.

7. The proposed method introduces substantial algorithmic complexity through frequency decomposition at each training step, K-stage sequential diffusion processing, sample-specific SNR estimation, and additional scheduler parameter tuning beyond standard diffusion hyperparameters. Despite this added complexity, improvements on certain datasets prove marginal: Temperature shows an MSE reduction from 0.0035 to 0.0024—merely 0.0011 absolute improvement on already small errors—while Solar demonstrates a decrease from 0.444 to 0.354, again in a regime where baseline errors are acceptably low. This complexity-benefit imbalance raises critical questions for practitioners who must evaluate whether the engineering overhead, computational cost, and implementation complexity justify incremental gains that may fall within statistical noise.

8. Ablation studies (Appendix D.1) are conducted only on synthetic data, never validating on the main benchmarks whether multiple components genuinely help. Table 14's synthetic results paradoxically show that K=1 (MSE=0.0978) performs comparably to K=3 (MSE=0.0936), suggesting the multi-component decomposition may be unnecessary when residuals have minimal variation. The extreme sensitivity to component number K lacks practical resolution, as the proposed automatic selection method is never empirically validated.

9. Visual results in Appendix G contradict performance claims: SSSD+FFT fails to capture initial dynamics, and Sashimi+FFT shows no visible improvement over baselines (scale is different so it is difficult to compare fairly).

10. Results are inconsistent—with unexplained performance degradation on ETTm1 (DiffWave: 1.093 to 1.774 MSE) and SSSD on Electricity (0.957 to 1.283)—yet these results receive no investigation.

11. Computational analyses are missing. The authors do not report memory complexity or FLOPs comparison, nor an inference time evaluation, despite 200-step sampling is provided.

12. Minor point: The paper also writes that regular diffusion adds noise "indiscriminately". Well, that is the point of diffusion models, that small noise denoising at every step is learnable. If indeed we impose structure, then other mechanisms might be more beneficial and diffusion is unnecessary!

**Questions:**

1. By isolating components during diffusion, does FFT-DM fail to model cross frequency effects? Could you provide: (1) theoretical or empirical analysis of what information is lost by assuming component independence, (2) ablation studies comparing joint versus sequential diffusion of components, and (3) examples where cross-frequency coupling matters for forecast accuracy? This would clarify whether the decomposition assumption introduces systematic biases that limit the method's applicability.

2. Could you provide a systematic analysis of when and why FFT-DM hurts performance? Specifically: (1) Is there a quantitative measure of dataset 'periodicity strength' that predicts whether your method will help or harm? (2) Do certain architectures (e.g., U-Nets like Sashimi) fundamentally conflict with staged diffusion? (3) What diagnostic tests should practitioners run before adopting FFT-DM to avoid these failure modes?

3. Could you provide direct empirical comparisons showing that integrating decomposition into the diffusion process outperforms applying decomposition in simpler, non-generative architectures?

---

### Official Review · Reviewer_4CZh · 2025-10-30

**Soundness:** 3
**Presentation:** 2
**Contribution:** 2
**Rating:** 4
**Confidence:** 4

**Summary:**

This paper proposes FFT-DM, a model-agnostic modification to the forward diffusion process in diffusion-based time-series forecasting models. Instead of injecting Gaussian noise uniformly across the entire signal, the authors first decompose the input series into frequency components using the Fast Fourier Transform (FFT), then apply noise sequentially according to each component’s energy. This design preserves dominant temporal patterns—particularly seasonality—throughout the diffusion trajectory. The approach does not alter the network architecture and can be integrated with existing diffusion models (e.g., DiffWave, CSDI, SSSD, Sashimi). Experiments across six benchmark datasets (Electricity, ETTm1, Temperature, PTB-XL, MuJoCo, Solar) demonstrate consistent forecasting improvements, especially for data with strong periodic patterns, with less than 7% computational overhead.

**Strengths:**

1. The paper rethinks the forward diffusion process instead of modifying architectures. By decomposing time-series signals in the frequency domain before noise injection, it captures temporal dependencies and preserves periodic patterns.
2. The staged forward process and SNR-based noise rescaling are mathematically grounded, offering an intuitive yet principled explanation of how noise interacts with temporal structures.
3. The proposed FFT-DM framework is model-agnostic and plug-and-play, easily integrated into existing diffusion models such as DiffWave, CSDI, and Sashimi with minimal modifications.
4. The method consistently improves stability, accuracy, and interpretability across diverse benchmarks—especially for datasets with strong seasonal or periodic behaviors—while incurring negligible computational cost.

**Weaknesses:**

1. The FFT-DM framework is mainly effective for periodic or quasi-stationary time series, limiting its applicability to datasets lacking clear frequency structures.
2. The method assumes the presence of meaningful frequency components, which may not hold for highly non-stationary or chaotic data (e.g., MuJoCo), reducing its advantage in such scenarios.
3. The performance heavily depends on the manually tuned number of frequency components 𝐾, which affects both accuracy and stability.
4. This sensitivity to 𝐾 raises concerns about the method’s robustness and generalizability across datasets with diverse spectral properties.

**Questions:**

1. Both this work and Diffusion-TS utilize FFT-based signal decomposition within diffusion models. Could you clarify how FFT-DM differs from Diffusion-TS in terms of methodology and design objectives, and what specific advantages FFT-DM provides?
2. During inference, how stable is the SNR estimation when the conditioning context is noisy or non-stationary?
3. Have you explored applying FFT-DM to probabilistic imputation or generation tasks beyond forecasting?

---

### Official Review · Reviewer_891o · 2025-10-31

**Soundness:** 3
**Presentation:** 3
**Contribution:** 2
**Rating:** 4
**Confidence:** 3

**Summary:**

This article proposes an adjusted forward diffusion process to address the characteristics of time series.

**Strengths:**

1. Strong performance: Perform well in multiple datasets.

2. Lightweight calculation: Little additional computational burden

3. Model independent: can be combined with existing method

**Weaknesses:**

1. The existing work that adjusted the adding noise process of diffusion in time series have not been thoroughly taken into account, such as [1]

[1] Wang C, Yang L, Wang Z, et al. A Non-isotropic Time Series Diffusion Model with Moving Average Transitions[C]//Forty-second International Conference on Machine Learning.

2. No standard deviation reported for the experiment. Has the author conducted multiple experiments?

3. The method of the paper seems to rely on the characteristics of the time series data itself, in other words, on the applicability of FFT to the time series while it doesn't demonstrate reliability in highly non-stationary data.

4. In Table 14, the true components of the sequence and the components that achieve the best results are not completely consistent, which means that the model does not have sufficient fitting ability. I am concerned that the performance improvement of this method may come more from more diffusion steps, etc.

**Questions:**

1. After dividing the components, is there a 200 step diffusion for each component, or the total diffusion step for all components is 200? If the latter, how to allocate to each component？

2. How does your model handle non-stationary time series with primary frequencies varying over time? Will the decomposition based on global FFT mislead such data?

3. During prediction, the model relies on historical data to estimate the future signal energy distribution $d_k$. What will happen if there is a sudden change in the time series data? Will it lead to a huge drop in performance compared to other existing method?

4. The author claims that the framework is decomposer independent, but only verified using FFT. Have you considered using decomposition methods such as wavelet transform that are more suitable for handling local and instantaneous features ?

5. When dealing with multivariate time series, should FFT decomposition be performed separately for each channel to determine different K values, or should a unified K be used? If it is the former, how to synchronize the staged diffusion processes of different channels?

---

### Official Review · Reviewer_EQPN · 2025-11-03

**Soundness:** 1
**Presentation:** 2
**Contribution:** 1
**Rating:** 0
**Confidence:** 5

**Summary:**

The paper proposes FFT-DM, a model-agnostic forward diffusion process for time-series forecasting that leverages Fourier decomposition to preserve structured temporal patterns such as seasonality. Instead of modifying the network architecture, FFT-DM stages noise injection according to the energy of spectral components, ensuring that dominant frequencies maintain higher signal-to-noise ratios throughout the diffusion trajectory. This design aims to delay the destruction of salient temporal structures during the forward process, thereby improving long-horizon forecasting. The method is evaluated on standard benchmarks (Electricity, ETTm1, PTB-XL, MuJoCo, Temperature, Solar), showing performance gains over several diffusion baselines and competitive results against Sashimi, with only ~7% computational overhead.

**Strengths:**

1. The idea of structuring the forward diffusion process according to spectral energy is conceptually sound and aligns with well-established time-series decomposition principles.

2. Empirical results demonstrate consistent improvements on datasets with strong seasonality, validating the core hypothesis in practical settings.

**Weaknesses:**

1. The theoretical foundation is shallow: the staged diffusion process lacks rigorous justification beyond heuristic signal-to-noise ratio (SNR) scaling, and the paper provides no convergence or stability guarantees for the modified forward/reverse dynamics.

2. The claimed ``model-agnostic'' nature is misleading, integrating stage index k into the denoising network (via input embedding) implicitly requires backbone modifications, contradicting the agnosticism assertion.

3. The method closely resembles prior frequency-aware or multi-stage diffusion frameworks (e.g., mrDiff, f-DM, Spectral Diffusion), yet the novelty is overstated; FFT-DM is essentially a recombination of known ideas (Fourier decomposition + staged denoising) without fundamental algorithmic innovation.

4. The SNR rescaling factor $d_k$ is estimated from conditioning data during inference, which assumes stationarity and may fail under distribution shift, a critical flaw for real-world forecasting where future dynamics often diverge from historical patterns.

5. Experimental comparisons are selective: Sashimi, a strong non-diffusion baseline, is used inconsistently (FFT-DM hurts Sashimi on PTB-XL), yet the paper omits ablation against simpler decomposition strategies (e.g., moving average) that might achieve similar gains with less complexity.

6. The paper ignores the ambiguity in component ordering: sorting by amplitude contradicts standard practice in time-series modeling, where low-frequency (trend) components are often more predictive than high-amplitude high-frequency noise.

7. Typos and notational sloppiness abound: e.g., ''chaUnlike'' (p.3), undefined symbols (τ=T/K without justification), inconsistent use of $x_0$ vs. $z_0$, and missing spaces in mathematical expressions (e.g., ``N(0,I)'').

**Questions:**

Please refer to Weaknesses 1–7 for specific technical and methodological concerns requiring clarification or correction.

---

### Author Response · Authors · 2025-12-02
**Withdrawal Notice**

We would like to thank the reviewers for the time and effort dedicated to evaluating our submission, as well as for the insightful comments provided. We have completed all suggested ablations, including wavelet and the hyperparameter sensitivity ablations,  and were preparing to submit a detailed, point-by-point response once the final results were ready.

 However, given the recent leak of author and reviewer identities and the subsequent handling of the situation, we believe it is best to withdraw our submission from this years ICLR.
The lack of a discussion phase with reviewers, combined with the presence of an outlier score of zero, despite raising points that are similar to other reviewers but with an evaluation that appears disproportionately severe, indicate to us that is better to withdraw from this venue, for this year.

Before closing, we would like to address what we found to be one of the most interesting remarks:

Minor point: *The paper writes that regular diffusion adds noise ˜indiscriminately". But that is the point of diffusion models, the denoising of small noise at each step is learnable. If we impose structure, then other mechanisms might be more beneficial and diffusion unnecessary.*

We respectfully maintain that incorporating structured constraints during the diffusion process is a promising direction. Having a multistep inference process is asking, in our opinion, to inject restrains and data/domain specific knowledge.
Just applying gaussian noise is the simplest approach, and it yields great results for image generation, but if we consider some information to be more valuable/meaningful than other during the diffusion process, in our case, low-frequency signals, the vanilla DDPM is not sufficient. This doesn't mean that diffusion is unnecessary, is just means that it can be expanded to include the advances and knowledge that already exists in time-series processing, particularly for tasks such as forecasting.

Prior work, such as ControlNet and the Reflected Schrodinger Bridge for constrained generative modeling, illustrates the value of guiding or restricting generative dynamics, and other works replace gaussian noise with masking or blurring, like soft diffusion, showing in a provable way that the deterministic part of the diffusion process can be modified.

We appreciate the reviewers efforts and thank the program committee for their work during this challenging review cycle.

---

### Note · Authors · 2025-12-02

I have read and agree with the venue's withdrawal policy on behalf of myself and my co-authors.